# Social Support and Acculturative Stress of International Students

**DOI:** 10.3390/ijerph19116568

**Published:** 2022-05-27

**Authors:** Ika Febrian Kristiana, Nugraha Arif Karyanta, Ermida Simanjuntak, Unika Prihatsanti, Tri Muji Ingarianti, Muhammad Shohib

**Affiliations:** 1Faculty of Psychology, Diponegoro University, Semarang 50275, Indonesia; unik0206@gmail.com; 2Doctoral Program of Psychology, Airlangga University, Surabaya 60286, Indonesia; optimissaja@gmail.com (N.A.K.); tri_ingarianti@yahoo.com (T.M.I.); m.shohib.umm@gmail.com (M.S.); 3Faculty of Psychology, Widya Mandala Catholic University Surabaya, Surabaya 60265, Indonesia; mida@ukwms.ac.id

**Keywords:** stress acculturative, social support, international students, meta-analysis

## Abstract

Continuing to study abroad brings challenges, apart from academic demands. International students are prone to acculturation stress as a consequence of cultural differences. Many research reports show that social support is a great buffer against experienced stress, but there has thus far been no study that analyzes the real effect of social support on acculturation stress. This meta-analysis study aims to investigate the true effect of social support on acculturation stress of international students based on studies reporting it. A meta-analysis was performed following PRISMA. The electronic databases used were Science-Direct, ERIC, ProQuest, Google Scholar (only for ETD), and opengrey.edu, with the article year limitations being 2009–2019. Eight (8) studies were involved in the meta-analysis. There were three instruments of acculturation stress and five instruments of social support that were used in the different studies. The effect size analysis showed that there was no difference in the effects of eight studies (z = −0.553; SE = 0.497; 95% CI = −1.248–−0.699; *p* = 0.580). Furthermore, there was no statistically significant moderator variable, the instruments used were quite diverse. The role of a moderator, other than gender, was not used because of limited information from the studies used. Social support plays a major role in reducing acculturation stress in international students. However, studies involving moderator and confounding roles need to be conducted.

## 1. Introduction

The number of international students worldwide reached 5.1 million, a significant increase from only 2.1 million in 2000 [1]. The number of students leaving home to study at overseas universities, including students from developed and developing countries, is expected to continue to increase every year [2]. There is even a possibility of exceeding seven million by 2025 [3]. The increasing number of students studying abroad cannot be separated from the positive impact gained on students’ cross-cultural understanding, their adaptation to cultural differences, and their foreign language skills [4,5]. In the long term, these experiences can help students find and do better jobs, and consequently improve their overall living conditions [6,7,8].

However, studying in another country requires more effort for most students. Apart from having to deal with academic work that is more challenging than in the previous education phase, international students also have to adapt to a new environment, even to a new culture, that may be completely different from their home region. Challenges in adapting to academic and social environments may include difficulties with English and communication, developing friendships, and lack of knowledge of the culture of the country of study [9], along with changes in food, finance, housing, and social support [10]. In addition, international students often experience higher levels of discrimination and homesickness than students from host countries [11]. This experience relates to the challenges and pressures involved during the acculturation process and in adapting to a new culture [10]. Some research shows how the process of adapting to a new place is a strong form of stress [12] and affects both physical and mental health [13,14].

Various literature has shown that maladjustment in international students negatively impacts psychosocial development and leads to a higher tendency towards mental health issues such as depression [15]. The new environment and cultural differences between home culture and host-culture experienced by students have the potential to cause obstacles, one of which is acculturative stress—stress that occurs as a result of the acculturation process [16]. Stress occurs when the experience of acculturation causes problems for the individual [17].

Various studies show the consequences of acculturation stress, including feelings of loss of certain behaviors that must be done differently or even cannot be done, as well as routines, habits, and intimate relationships that must also change [9]. Several other researchers have revealed that various forms of reactions to acculturation stress can include anxiety, stress, frustration, fear, and pessimism; feelings of alienation; and racial discrimination [15,18,19,20].

Studies on acculturation stress, especially on international students, have been widely carried out, mostly on subjects in the North American region [21], especially in the community and students of Latina/Latino and international students from China. Various studies show predictors of acculturation stress, such as age, continental origin, English language ability, time/length of stay in the host-culture, travel experience [22], socio-cultural adaptation, comfort using language and gender [23], type of acculturation, age of respondent, age at migration, generation status, income of parents and students, social support, and English proficiency [24,25], as well as identification of the host culture and preservation of indigenous culture [26].

A longitudinal study of international students showed a reduction in acculturative stress and homesickness during the last week of the first year compared to the first week of the first semester. Finally, acculturative stress at the beginning of the first year and satisfaction with the college experience at the end of the first year mutually influence each other over one year [27].

Studies on acculturation stress also discuss the impact of exposure to stress, for example on quality of life and psychological health regarding its impact on academic achievement (e.g., [24,28,29,30,31,32]), although several researchers have found different effects.

The literature on stress-coping shows that social support is an effective buffer for stress [33]. Various studies have also shown that various forms of social support can mitigate the level of acculturative stress experienced by international students [19,25,34].

Several different studies have shown the importance of social support from local friends for international students to reduce homesickness [35,36]. Friendships with local residents are important to develop cultural knowledge and competencies needed in adaptation, while social support from family and fellow countrymen will help students to maintain cultural identities and practices, and reduce homesickness and disorientation [37]. However, different results are also shown by several studies, which concluded that social support for international students does not have a direct effect on negative events experienced by students, such as racism and psychological distress symptoms [14,15,38].

There have been many studies on the effect of social support on acculturation stress with international student participants. However, so far, there has been no systematic review and meta-analysis to clarify the effect of social support on acculturation stress on the international student population and what measures are most widely used to assess acculturation stress on this population. It is known that social support influences acculturation stress, but it is also important to know for sure how strong the effect of social support is at minimizing acculturation stress experienced by international students with the appropriate (valid) instrument. Thus, this will need to be taken into account for formulating interventions or policies for universities that utilize social support in minimizing the acculturation stress of foreign students.

Thus, this study aims to conduct a systematic review and meta-analysis of cross-sectional studies on the effect of social support on acculturation stress on international students. Specifically, the objectives are (1) to test the magnitude of the effect (influence) of social support on international students; (2) to determine which types of social support and acculturation stress are included in the study; and (3) to identify potential moderators of acculturation stress in international students, for example sociodemographic variables.

## 2. Materials and Methods

This systematic review and meta-analysis were carried out following a protocol based on the PRISMA statement [39]. Meta-analysis is also possible in various observational studies because bias is more likely in observational studies. With certain analytical techniques in the meta-analysis, we will be able to see the cause of the bias in observational studies (non-clinical). Viewed from this process, meta-analysis is a study retrospective observational studies, in the sense that the researcher makes a recapitulation of facts without performing experimental manipulations. Furthermore, a meta-analysis conducted after a systematic review will potentially obtain accurate information about the effect of a variable and/or treatment from quality research. The references are as follows.

### 2.1. Eligibility Criteria

The PICO approach to cross-sectional studies, as defined in the PRISMA Guidelines [40], was implemented. The criteria for the inclusion of research or studies were as follows:

#### 2.1.1. Participants’ Characteristics

Studies were included if they were carried out on international students from countries with individualistic or independent values who were studying in countries with collectivism values or vice versa. 

#### 2.1.2. Result Characteristics

Studies were included if both social support and acculturation stress instruments were used with known psychometric properties. Studies were included only if the results examined and reported the effect value (*r*, *t*, or *F*) of the social support variables, both unidimensional and multidimensional (using the sub-scale), including support from family, support from old friends, support from new friends, or support from study places, on the variable of acculturation stress, both unidimensional and multidimensional (using the subscale), including homesickness, fear, guilt, culture shock, perceived hate, and perceived discrimination. 

#### 2.1.3. Design Characteristics

Studies were included if they used a cross-sectional or randomized control design that examined the effect of social support on acculturation stress on international students from countries with individualistic cultures to countries with collectivistic cultures, or vice versa, otherwise they were not included. 

### 2.2. Information Sources and Search Strategies

Sources of information included the following databases: ScienceDirect, ERIC, proquest, Google Scholar (only for electronic thesis and dissertation/ETD), Indonesian repository (RAMA Repository or RIN Repository), and gray literature (opengrey.edu). The article/data year were limited to the last 10 years (2009–2019), and only articles in English or Indonesian were included. The search was conducted for 1 month, from 4 November to 4 December 2019. The search was conducted using the following keywords, or a combination (“social support” OR “family support” OR “old friend support” OR “new friend support” OR “university support” AND “stress acculturative” OR “homesickness” OR “fear” OR “guilt” OR “culture shock” OR “perceived hate” OR “perceived discrimination”).

### 2.3. Data Collection Process

Five authors (E.S., U.P., T.M.I., N.A.K. and M.S.) conducted a literature search, then assessed the eligibility criteria for the articles. At this stage, the study was screened regarding the inclusion criteria by reading the title and abstract. In the next stage, the remaining studies were screened using the eligibility criteria after reading the full text. After data collection and extraction, one author (I.F.K) compared the results to reach a final consensus based on the inclusion and exclusion criteria, this process also involved two independent reviewers (F.N.M. and T.M.S.) to obtain a collection of studies included in the meta-analysis. After this stage was carried out, data were collected based on the following characteristics in each article: author, year of publication, research design, sample size, age and gender, and the variables studied. 

### 2.4. Risk of Bias in Individual Studies

The main potential confounding bias in the study was identified, with a focus on the following: measurement bias, lack of accurate operational definitions of both social support and acculturation stress, and selection bias (e.g., how to obtain research samples not using probability sampling). 

### 2.5. Assessment of Study Quality

The assessment of the study quality was carried out by examining the study details using a correlational study checklist developed and adapted by [41]. The developed checklist provides 15 items that assess various features of the research methodology, including reporting on the validity and reliability of measuring instruments, number of participants, reporting of participant characteristics (for example mean and SD age, and gender ratio), reporting effect size (raw *r* value), selection to avoid bias, are there any missing or drop out data, control of confounding variables, analysis with appropriate interpretation, appropriate statistics, and whether any participants are vulnerable.

### 2.6. Moderator Coding

When the inconsistency analysis showed a large and significant heterogeneity between the effect sizes, the role of the moderator was investigated, and two independent reviewers (A.P. and C.J.) coded the moderator, extracted data from the primary study, and entered it into an excel spreadsheet. Any potential differences were discussed and resolved with the third reviewer (J.M.). The variable gender sample was coded as a representation of a higher percentage of female participants

### 2.7. Data Extraction and Summary

Heterogeneity in the studies, including the effect sizes, was calculated using the randomized effects model. Data requested for the effect size calculations were extracted independently by two meta-analysts (A.P. and C.J.). Effect sizes were estimated using 95% confidence intervals and were interpreted according to the criteria suggested by [42]. Thus, an effect size of 0.80 or more was assumed to be large, 0.50 moderate, and 0.20 small. The global effect size was calculated as the average effect size obtained by combining the effect sizes associated with the forms of social support obtained and the acculturative stress experienced. Mean effect sizes were calculated for each study as a result of social support or global acculturation stress.

### 2.8. Inconsistency Analysis

To assess the heterogeneity between studies, two complementary indicators, I2 index [43] and Q statistic [44], were used. Values of near zero indicate homogeneity, while values of 25–50%, 50–75%, and 75–100% represented low, moderate, and great heterogeneity, respectively [43].

### 2.9. Moderator Analysis

Given that the inconsistency analysis suggests great heterogeneity, the moderator analysis mentioned above was performed using a mixed-ANOVA model and meta least squares regression or meta regression. Meta regression was performed by adding the participants’ vulnerabilities (by coding the number of women).

### 2.10. Publication Bias

To investigate the possibility that effect sizes were subject to publication bias, Duval and Tweedie’s trim and fill procedures [45] and visual inspection of funnel plots were used to check them.

## 3. Results

The results of the study review in this paper will be described as follows.

### 3.1. Selection and Descriptive Characteristics of the Study

The electronic search produced 49 records after the duplicates were removed. Of these studies, 18 were excluded on titles or abstracts because the construction focus was unrelated to the aims of this study. Thus, 31 studies were screened on the full text for inclusion. From these studies, 17 studies were excluded, in which five studies did not really examine the social support or could be said to examine the constructs of the variables adjacent to the social support variables, for example social interaction or social connectedness, while the other 12 studies did not use cross-sectional or randomized control designs. Therefore, we obtained 10 studies screened for systematic review and meta-analysis. Furthermore, two other studies were excluded because they did not report sufficient data to calculate the effect sizes (e.g., standard deviation of quality of life or sampling variance), and the authors did not respond when they were contacted for the required data. An overview of the study implementation procedure is provided in the flowchart in Figure 1.

Eight studies were included in the meta-analysis process, one of which was a thesis, while seven studies were included in a peer-reviewed journal. Five studies were conducted in the United States, one in Malaysia, one in Cyprus, and one in Korea. The total sample size ranged from 100 to 272 participants. The study publication years ranged from 2010 to 2019. Regarding the instruments used for social support, three studies [35,46,47] used the Index of Sojourner Social Support (ISSS), two studies [48,49] used the Multidimensional Scale for Perceived Social Support (MSPSS), one study [50] used the Index of Social Support (ISS), one study [51] used Perceived Support from School, and one study [52] used the Interpersonal Support Evaluation List. Whereas for the acculturation stress measurement instrument, five studies [46,47,48,51,52] used the Acculturative Stress Scale for International Students (ASSIS), two studies [35,50] used the Index of Life Stress (ILS), and one study [49] used the Acculturative Stress Scale for Chinese Students (ASSCS). For further details, the characteristics of the studies included in the meta-analysis can be seen in Table 1.

### 3.2. The Results of Statistical Test for Meta-Analysis Study

The index was close to zero, indicating homogeneity [43].

#### 3.2.1. Inconsistency Analysis

To assess the heterogeneity of the studies, a random effect model (REM) estimation was carried out by looking at the I^2^ index [55] and Q statistics [44,56] (Table 2a,b).

Based on the output table above, the values of I^2^ = 93.36% and Q = 105,454 were very high, which means that they indicated a low homogeneity or the effect size values were too varied. 

Based on Figure 2, from the comparison forest plot of the studies, the pooled effect size was −0.34, but this does not mean anything because of the high of I^2^ values. Thus, it is necessary to find the cause of the heterogeneity by doing a meta-regression or moderator analysis.

#### 3.2.2. Moderator Analysis

Given that the inconsistency analysis suggests great heterogeneity, a moderator analysis, mentioned above, was conducted using a mixed-effect ANOVA-mixed model and meta-least squares regression (meta-regression). Meta-regression was performed by adding the participants’ vulnerabilities (by coding the number of women). The following is the meta-regression output (Table 3a,b).

Based on the output above, the index I^2^ experienced an increase while the Q statistic value decreased, even though it was a little from the REM analysis, thus the vulnerability of participants was not a significant variable in determining the heterogeneity of the effects.

#### 3.2.3. Publication Bias

To test the bias publications, we used the funnel plot technique. If it was not symmetrical using the Egger’s test (*p* < 0.05), there was publication bias. Studies with accurate effect size estimations are located in the top of the curve.

Based on Figure 3, the funnel plot shows an asymmetrical shape. The regression value (z = 1.325; *p* = 0.185) and the rank correlation test for plot asymmetry were also insignificant (*p* = 0.275), indicating that the results were unlikely to be due to publication bias. With meta-regression, the five-funnel plot image remained symmetrical, but the dots were at the bottom, which indicates inaccurate estimation.

## 4. Discussion

Adjustment to the campus environment includes several aspects, namely adjustment with friends, adjustment to courses, adjustment with lecturers, adjustment with employees, and adjustment to the campus organization [15]. Changes in the environment or conditions encountered when becoming a student require an appropriate adjustment process. Students who fail to make adjustments to the conditions will face various problems, which will eventually build their own pressure.

The changes they experience do not only concern everything related to campus conditions, such as academic demands, but also other conditions, such as leaving home and parents to live abroad, searching for identity, and making decisions related to career choices. The adjustment of foreign students includes not only the factors mentioned above, but also cultural factors, which make them vulnerable to stress [15,19,20].

Previous research has shown that foreign students or international students who have difficulty in adjusting experience anxiety and may suffer from depression [18,19]. This is consistent with the findings of a literature review, that foreign students experience problems related to their adjustment [46]. It can be an important consideration in the literature review used here that social support has a contribution to reducing acculturative stress.

From the eight (n = 8) studies included in the meta-analysis, it was found that there was a large heterogeneity between the studies. Publication bias was also identified from the meta-analysis, so it is difficult to say how accurate the effect size of this social support is at reducing acculturation stress in international students. Although gender vulnerability does not show its effect on the variation in the effects of social support on acculturation stress, several other things from their background characteristics may influence the variation in these effects, for example, length of stay in the study destination country, marital status, and multicultural experience. Unfortunately, of the eight studies involved in the review process, only one study provided detailed information on the characteristics of the length of stay of participants [52]; however, Kim and Yoo did not further analyze the effect of the differences in the length of stay of participants in the study destination country with the acculturation stress level. Previous research has reported that length of stay is a factor that also affects international student stress (see [57,58]). The adjustment process of international students studying and living in a foreign country begins with a joyful process when students first arrive in a new culture; the difference is interesting and attractive, and students are stimulated and curious. This stage is called the “honeymoon” stage [58]. After some time, the excitement of the first stage may gradually fade away. Individuals may feel isolated due to cultural differences, and feel the absence of family or friend support. This is called the “distress stage”.

Meanwhile, the eight existing studies did not provide detailed information about marital status and previous multicultural experiences. Based on previous research, it is believed that international students who have spouses in their home countries may feel more stressed due to long-term separation [59,60]. Multicultural experiences such as intercultural course learning have an effect on acculturation adaptation, which can further minimize the risk of acculturation stress [61,62], but other forms of multicultural experience, for example having lived abroad or having a multicultural family background, have not been found to have an effect on acculturation stress of international students.

Furthermore, the studies involved in this systematic review and meta-analysis did not specifically examine the differences in cultural values between the destination countries for international students, as Hofstede’s model of cultural theory explains that the cultural dimension describes the effect of a society’s culture on the values of its members, and how those values relate to behavior [63]. For example, individualism vs. socialism, which describes the degree to which individuals are integrated into groups. In individualistic societies, stress is placed on personal achievement and individual rights. People are expected to stand up for themselves and their immediate families, and to choose their own affiliations. By contrast, in collectivist societies, individuals act primarily as lifelong and cohesive members of groups or organizations (note: “The word collectivism in this sense has no political meaning: it refers to the group, not to the state”). Some people have large families, which are used as protection in exchange for unquestioned loyalty. Regarding the individualism index, there is a clear gap between developed and Western countries on the one hand, and less developed and Eastern countries on the other. North America and Europe can be considered as individualists with relatively high scores. In contrast, Asia, Africa, and Latin America have strong collective values. This finding is also a recommendation for further studies, that paying close attention to differences in cultural values, especially in cross-cultural studies, is very important.

The variety of studies that report on the size of the effect of social support on acculturation stress on international students opens up wide opportunities for future research, especially research with more rigorous methods, in order to examine the effect of social support on acculturation stress and take into account some of the recommendations of this study. However, although the results of the meta-analysis indicate publication bias and inaccuracy of effect sizes, at least the information that is available shows that there is a relationship between social support and acculturation stress of international students, and is an important consideration in the development of various programs and related policies.

An interesting finding is that gender, in the context of this meta-analysis, namely women as vulnerable participants, is not a factor influencing heterogeneity between studies in the meta-analysis. One thing that needs to be considered is the culture of the country of the study destination. This can be seen from the study by [48], which was conducted in Asian countries, which found that acculturative stress was negatively related to social support in a sample of international students who were studying in Malaysia. [52] Found similar results, that social support is negatively correlated with acculturative stress in international students in Korea, which was similar to the results of research conducted in the United States [35,46,47,49,50,51]. The largest contribution value (*r*) was found in the study of [48], within the context of Malaysian culture. Collective culture makes individuals feel like being part of the group, so that when they are in a situation outside their group, they feel anxious [15]. Thus, in countries that have strong cultural values of collectivity, the role of social support will be significant for the acculturative stress felt by international students.

These studies are in line with the previous studies in different contexts, that social support produces positive psychological outcomes such as reduced stress levels [64]. Research has shown that social support is an important source of coping with stress. In order to get social support, international students will always be in contact with friends and family through social media [15]. This will feel more comfortable, because being in a new country tends to force students to be limited in expressing what they feel in a foreign language [35]. With the results of this literature review, it is evident that individuals who have high social support tend to experience low stress.

In connection with the results of the research on the significant effect of social support in reducing acculturation stress, practical suggestions to facilitate the student adjustment process include that universities and institutions deal with international students by considering the importance of providing a socially supportive environment. This can also be done by providing introductory activities that can provide information about the culture of the country of the study destination, especially with regard to the culture or academic and social life of international students, providing homestay or foster parents opportunities, and designing multicultural learning and extra activities. Students themselves are expected to play an active role in obtaining social support, for example to involve themselves in organizational activities that will facilitate meeting other students on campus.

## 5. Conclusions

The results of the meta-analysis and systematic review showed there was a correlation, but an accurate effect size could not be determined, between social support and stress acculturative on international students. This is due to the high degree of heterogeneity and indications of publication bias. The limitation of this study is that not all studies have fully reported matters related to the effect size value, such as the sampling variance value, which is not always included in existing studies. In the subsequent meta-analysis studies, there were more studies conducted on students that were not listed online, which could help to obtain more accurate results to describe the relationship between social support and acculturative stress.

## Figures and Tables

**Figure 1 ijerph-19-06568-f001:**
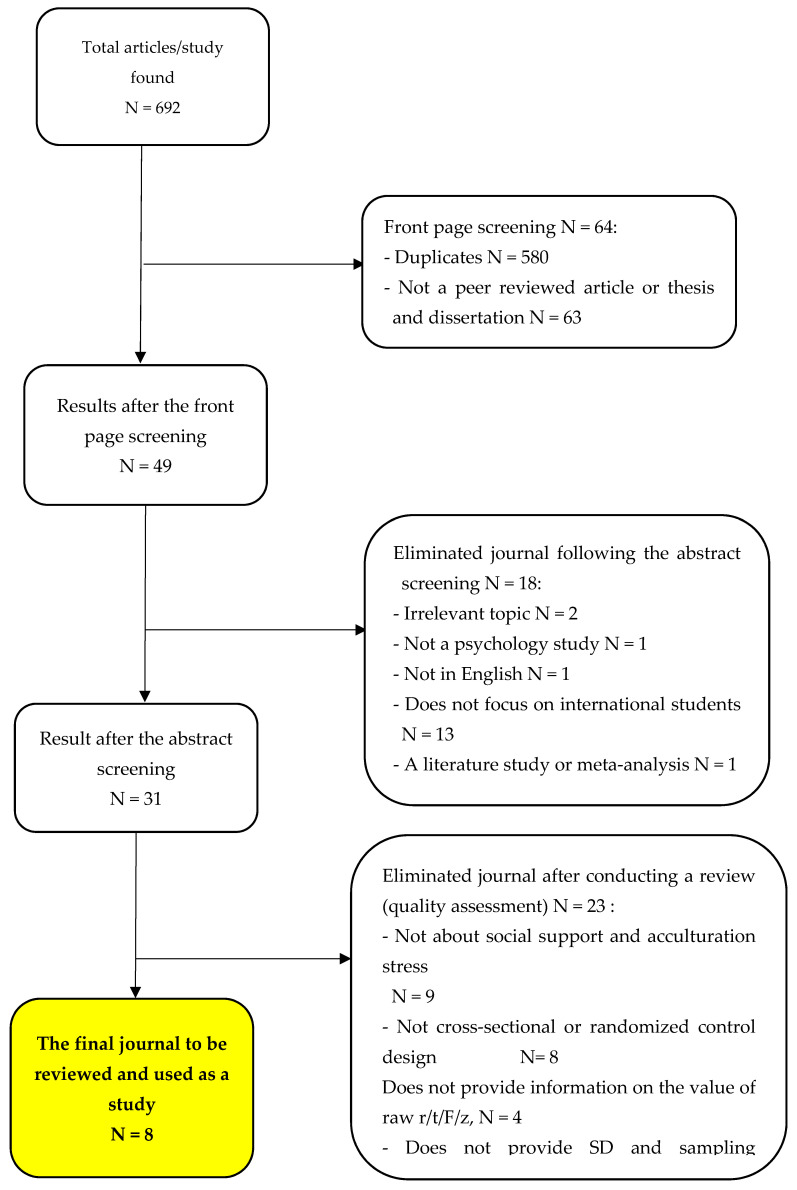
PRISMA flowchart study selection.

**Figure 2 ijerph-19-06568-f002:**
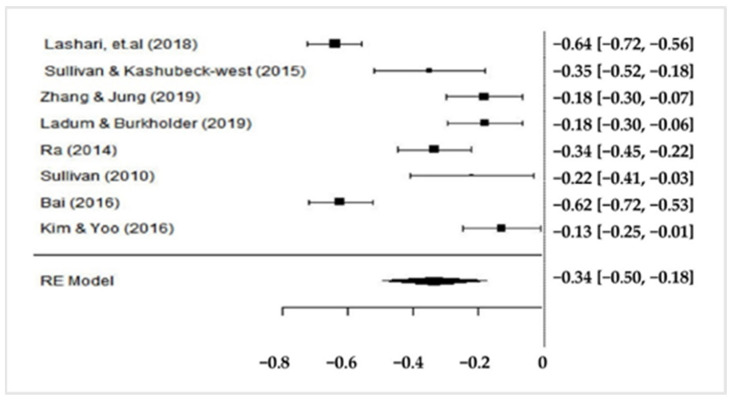
Forest plots comparison of the studies in the meta-analysis.

**Figure 3 ijerph-19-06568-f003:**
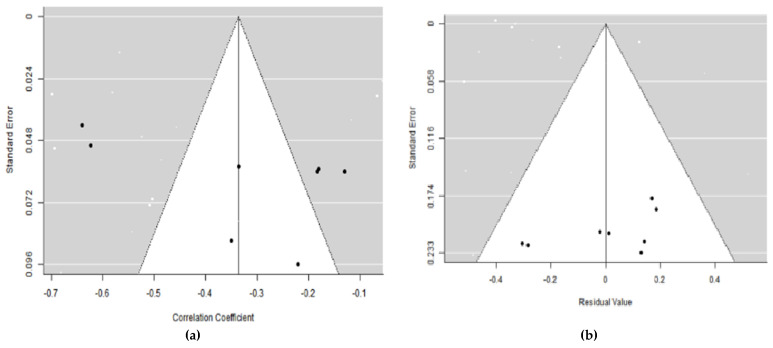
(**a**,**b**) Funnel plot with REM and mixed effect.

**Table 1 ijerph-19-06568-t001:** Description of study characteristics in the meta-analysis (n-8).

Study Name	Publication Date	Country	Sample Age and Gender	*n*	Instruments Used	Outcome
Lashari, et al.	2018	Malaysia	Gender ratio m:fm (55:45); Mean age = NA; SD age = 7.07	200 from 3 state university in Malaysia	Acculturative Stress Scale for International Students (ASSIS) α = 0.94 Multidimensional Scale for Perceived Social Support (MSPSS) α = 0.93	Acculturation stress was negatively related to social support (*r* = −0.64, *p* < 0.001)
Sullivan, et al.	2015	US	gender ratio m:fm (37.5:61.5); Mean Age = 25.43; SD Age = 5.23	104 from various countries studying in medium street public urban Univ. Midwest US	Acculturative Stress Scale for International Students (ASSIS) α = 0.94 Index of Sojourner Social Support (ISSS) α = 0.96	Social support from host nationals was negatively related with reported level of acculturative stress (*r* = −0.35, *p* < 0.01)
Zhang, et al.	2018	US	Mean age = 23; SD age = 2.8	137, Chinese international student di Northeastern US University	Acculturative Stress Scale for Chinese Students (ASSCS) α = 0.94 The Multidimensional Scale of Perceived Social Support (MSPSS) α = 0.91	Perceived social support from family was positively associated with students’ homesickness (*r* = 0.19, *p* < 0.01), perceived social support from friends was negatively associated with students’ perceived discrimination (*r* = −0.17, *p* < 0.05), perceived social support from school was negatively associated with students’ perceived discrimination (*r* = −0.23, *p* < 0.01), fearfulness (r = −0.20, *p* < 0.01), stress due to change (*r* = −0.15, *p* < 0.05) and guilt (*r* = −0.21, *p* < 0.01) – mean of school support = −0.19.
Ladum, et al.	2019	Ciprus	Mean age = 22.20, SD Age = 2.41	271 undergraduate students from 25 countries studying in English-medium programs at a university in the northern part of Cyprus.	Acculturative stress was measured using the Acculturative Stress Scale for International Students (ASSIS) [47]. α = 0.94. The Index of Sojourner Social Support (ISSS) Scale α = 0.87	Students with less social support experienced more acculturative stress (*r* = −0.18, *p* = 0.017)
Ra, et al.	2014	US	Mean Age = 24.48 SD Age = 4.53	232 East Asian international students from China, South Korea, Taiwan, and Japan who were registered in U.S. higher education institutions from 23 states across the United States	Index of Life Stress (ILS) α = 0.89 Index of Social Support (ISS) α = 0.94	Social support was significantly and negatively related with the outcome variable with a correlation of −0.336 (*p* < 0.0005). The correlations between acculturative stress and the three subscales of social support were −0.243 (*p* < 0.0005) for family dan old friend, −0.280 (*p* < 0.0005) for new friend in the US and −0.249 for universities and colleges.
Sullivan, et al.	2010	US	Gender ratio m:fm (52:48); mean age = 25.06; SD age = 5.069	100	Index of sojourners social support/ ISSS (reliabilities α each subscale: 0.95 home; 0.95 other international students; 0.94 host). Modifiying index of life stress test re-test *r* = 0.87; KR 20 = 0.86	Acculturative stress found to be negatively correlated with all three sources of social support: home (*r* = −0.15, *p* = 0.00), other international students (*r* = −0.18, *p* = 0.00) and host (*r* = −0.34, *p* = 0.00).
Bai, et al.	2016	US	NA	152	Acculturative Stress Scale for International Students (ASSIS) α = 0.949 Perceived Support from School. α = 0.789	Perceived support from school was significantly correlated with acculturative stress (*r* = −0.623)
Kim, et al.	2016	Korea	Gender ratio m:fm (92:180); mean age = 25.74; SD age = 2.68	272; Academic degree Undergraduate = 39, Masters student = 174, Doctorate student = 59, Length of Stay in Korea (months) 6–12 = 26, months 13–36 =108, months 37–60= 85, months > 61 = 53, Residence Dormitory = 98, Home stay = 132, Lease = 9, Others = 33	Interpersonal Support Evaluation List [53] (Cohen, S., & Hoberman, H. (1983) α = 0.83. Acculturative Stress Scale for international students [54] (Sandhu & Asrabadi, 1994) α = 0.84	Acculturative stress showed negative correlation with social support (*r* = −0.13)

Note: NA (not available); *n* (total participants); *r* (coefficient correlation); *p* (significant value).

**Table 2 ijerph-19-06568-t002:** (**a**,**b**) Heterogeneity test.

(**a**)							
	**Estimate**	** *se* **	** *z* **	** *p* **	**Cl Lower Bound**	**Cl Upper Bound**	
Intercept	−0.336	0.0817	−4.12	<0.0001	−0.496	−0.176	
(**b**)							
**Tau**	**Tau^2^**	**I^2^**	**H^2^**	**R^2^**	** *df* **	**Q**	** *p* **
0.222	0.0491 (SE = 0.0303)	93.36%	15.065	−	7	105.454	<0.0001

Note. Tau^2^ Estimator: DerSimonian-Laird.

**Table 3 ijerph-19-06568-t003:** (**a**,**b**) Mixed effect of the study review.

(**a**)							
	**Estimate**	** *se* **	** *z* **	** *p* **	**Cl Lower Bound**	**Cl Upper Bound**	
Intercept	−0.339	0.15392	−2.396	0.0166	−0.670	−0.067	
Moderator	3.89 × 10^−4^	0.0015	0.26	0.7952	−0.003	0.003	
(**b**)							
**Tau**	**Tau^2^**	**I^2^**	**H^2^**	**R^2^**	** *df* **	**Q**	** *p* **
0.236	0.0555 (SE = 0.0368)	94.16%	17.127	0%	7	102.764	<0.0001

Note. Tau^2^ Estimator: DerSimonian-Laird.

## Data Availability

The datasets generated for this study are available upon request from the corresponding author.

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
