# Peer review of "Social Support and Acculturative Stress of International Students"

_ijerph, 2022, doi:10.3390/ijerph19116568_

Round 1

Reviewer 1 Report

This article looks at the impact of study abroad on general mental health of participants.  The importance of understanding mental and social change as part of the study abroad program is a well researched topic.  The authors provide an interesting approach to reviewing the literature on study abroad and its impact on mental health.  I had several concerns with the article.  First, the literature review does not address several of the key issues of mental stress associated with the study abroad experience.  For example, the authors fail to address the importance of time in country as a critical influence on stress levels.  Many studies have assessed that the length of the study abroad experience strongly influences the level of stress.  In addition, family history is a critical element, as those with histories of intercultural encounters tend to adapt better in study abroad environments.  I would suggest the authors spend more time looking at the study abroad literature as it relates to stress and adaptation.  

Second, I'm not sure of the value of the statistical analyses in the article.  The article seems to be a literature review of the key issues that influence mental health in a study abroad environment.  This alone is a worthwhile research project that will help advance the academic discourse.  However, the authors provide little context and justification for the complex modeling they employ to seemingly reach the same conclusion one could reach through a through review of the academic literate.  In order for me to value the statistical modeling they used in the paper, they will need to explain what it adds to the academic literature, as opposed to a traditional review of the literature on the topic.  

Overall, I would ask you to address the deficiencies in the literature review and the justification in the model in the next draft of your research.  Also, please proofread your paper.  There are a few minor grammar, usage and word choice errors in the paper.  They are not significant, but they do detract from the readability of your draft.  

Thank you for the opportunity to review your research.  I think this is an interesting topic, one that has a niche within the academic literature.  Best of luck in the future.  

Author Response

Dear reviewer,

Thank you for the useful suggestions. We have revised several things based on your note of review: 

  1. Add an explanation of the length of stay and multicultural experience in the manuscript (see the discussion section)
  2. Do proofreading

However, we apologize that we cannot omit statistical analysis in the manuscript because we think it is quite clear that we have written them according to the meta-analysis design used. Thank you

regards,

Ika

Reviewer 2 Report

The paper deals with an interesting research topic in psychological field, i.e. the Social Support and Acculturative Stress, and shed a light on these constructs in international students.

This meta-analysis and systematic review is particularly interesting because it considers relevant dimensions of our world and in particular about the topic of internationalization and the link with the educational.  The abstract presents the rationale of the research and summarize the main findings.  

Introduction and discussion are generally well prepared and I think that the organization of these sections are good, but I have few suggestions to authors that might help them to improve and clarify the paper. 

More specifically, I think that a link to social support also in the educational field can be helpful to better guide the readers through the narrative. For example, a recent study by Lombardi et al., 2021 (A Comparison on Well-Being,Engagement and Perceived School Climate in High School Students with Learning Difficulties and Specific Learning Disorders), discusses this with regard to specific learning disorders and in particular to the peer group.  I would therefore suggest expanding and referring this type of literature (not in a broad way, certainly) in order to better capture it within the educational context. 

Please pay attention to tables (they are often cut off) and figures! 

Author Response

Dear reviewer,

Thank you for the very useful advice. We have tried to make revisions as you suggested, including:

  1. Follow the flow of writing from Lombardi et al (2021). We do it even though it's not the same because the problems and scopes are much different
  2. Do proofreading

Thank you

regards,

Ika

Round 2

Reviewer 1 Report

Thank for addressing my comments in your revision.  I appreciated the significant improvements to the literature review section of the paper.  However, the significance you place on individualistic versus collective cultural identity should be explained in the literature review, specifically Hofstede's work on dimensions of cultural difference would be an interesting framework to explain the challenges students face when studying abroad.  

I still have significant concerns on the relevance of the quantitative model in your paper.  I think your challenges in determining effect size is related to the ability of the model to explain your research question.  I think your question is important, well defined and is mostly explained by the literature review.  However, I think the quantitative model adds very little to your ability to answer the research question.  I stand by my previous suggestion that you remove the quantitative section of the paper and concentrate on a solid literary analysis of the 8 articles you identified as significant to your research question.  

Also, I would suggest another review of your style, grammar and usage in the paper.  Many of your sentences are awkwardly constructed and I am often confused as to their meaning.  

Overall, I really think the topic is useful, relevant and interesting.  I like where you are situating the research in the wider academic discourse.  I think the use of PRISMA protocol is appropriate, even though I would suggest you explain the protocol in more detail in the paper.  However, I do not think the quantitative model is appropriate for this paper.  

Best of luck in the future with your research.  I would suggest you keep working on the paper as it is a worthwhile topic.  

Author Response

Dear Reviewer,
Thank you very much for the very meaningful suggestions and input. The author has tried to add an explanation of the importance of meta-analysis as the method chosen to strengthen the reason for using this design. The authors also attempt to add discussion to the findings of the study associated with Hofstede's cultural model.
